# OpenReview forum: "FEABench: Evaluating Language Models on Real World Physics Reasoning Ability"
_ICLR.cc/2025/Conference — Submitted to ICLR 2025_

### Official Review · Reviewer_oeq1 · 2024-10-18

**Soundness:** 2
**Presentation:** 3
**Contribution:** 2
**Rating:** 5
**Confidence:** 4

**Summary:**

This paper introduces FEABench for evaluating LLMs ability on solving physics, mathematics and engineering problems using finite element analysis software. The authors evaluate 3 LLMs on this benchmark and also develop a multi-turn LLM agent that can interact with the FEA software API to iteratively improve its solutions. Experimental results demonstrate that this benchmark is challenging.

**Strengths:**

To the best of my knowledge, this is the first systematic work on large language models manipulating complex computer software.
The benchmark is challenging.

**Weaknesses:**

1. The FEABench Gold dataset is relatively too small, with only 15 problems. Is it comprehensive? Please compare with other relevant datasets to show whether your dataset is relatively too small, e,g, SWEBench[1] (Agent Dataset), OSWorld[2] (Software Agent Dataset).
2. The work is not particularly outstanding in the context of LLM-Agent research and lacks sufficient references to related work in this area. Although the narrative is from a physics perspective, the overall approach is closely related to LLM Agent research.
3. The experimental evaluation is not comprehensive enough. It only explores 3 closed-source models. Other models, including many open-source ones, e.g., Llama, were not tested.
4. The study only considers one software, COMSOL Multiphysics, which may not fully capture the "REAL WORLD PHYSICS REASONING ABILITY" claimed in the title, e.g. ANSYS.
5. The paper does not consider the graphical user interface of COMSOL Multiphysics, which is crucial for using this software, as is shown in Figure 5 in your paper.

[1] SWE-bench: Can Language Models Resolve Real-World GitHub Issues?
[2] OSWorld: Benchmarking Multimodal Agents for Open-Ended Tasks in Real Computer Environments

**Questions:**

Same as Weaknesses.

---

> ### Author Response · Authors · 2024-11-23
> **Response**
>
> We are grateful for your constructive feedback and have updated our work accordingly.
> *1\. The FEABench Gold ...*
> We acknowledge the concern about size and hope to address it in future work. FEABench Gold has 15 manually verified problems and 200 algorithmically parsed problems– FEABench Large. However, it can be challenging to scale verified problem generation in this domain. SWE-bench ([1]), when originally released, was an algorithmically parsed dataset with software engineering problems derived from Github issues. SWEBench-Verified, subsequently released by OpenAI earlier this year, consisted of 500 problems verified by a team of human annotators. The OSWorld dataset consisted of 369 manually annotated problems consisting of problems that involve interacting with everyday use apps on Operating Systems.
>
> *2\. The work is ...*
>
> Our primary objective behind the agent was to design the building blocks that would comprise an interactive LLM-Multiphysics Agentic interface, and to demonstrate a possible path toward enhancing the ability of the LLMs to generate code that solves such problems. The landscape of recent agentic research on code generation has largely targeted software development in languages such as Python that are well-represented in training data. SDE workflows often require agility and efficiency in non-trivial file navigation, a task that is not the most relevant skill to the workflow of an FEA engineer. As a concrete example, solving a problem in FEABench Gold does not require a user / LLM to navigate changes to pre-existing code across multiple modules. Even more involved FEA problems would at best involve loading parameter files and geometries files, neither of which resembles SWE workflows.
>
> **Revisions:** We acknowledge your observation and have better placed our work in the context of recent work devising agentic frameworks for general-purpose Python programming and further added references to related work in Section 5, Discussion and Related Work. We have also rewritten Section 5 (Agent-Multiphysics API Interface) to better clarify our focus on designing the building blocks of possible multiphysics reasoning agents as opposed to introducing an general-purpose agentic algorithm.
>
> *3\. The experimental evaluation...*
> We **added evaluations of Open Source LLMs and a Fine-Tuned LLM** and address this concern in Point 1 of the overall response.
>
> *4\. The study only ...*
> We hope we have addressed this concern in Point 2 of the overall response.
>
> *5\. The paper does ...*
> We address this comment in Point 3 of the overall response.
>
> We thank you for your time and look forward to addressing any other queries during the discussion phase.
>
> **References:**
> [1]: Jimenez et al 2023. SWE-bench: Can Language Models Resolve Real-World GitHub Issues? arXiv:2310.06770
> [2] https://openai.com/index/introducing-swe-bench-verified/

---

> ### Comment · Reviewer_oeq1 · 2024-11-23
>
> ## Agentic Research on Code Generation vs. LLM-Agent Work
>
> My concerns pertain to the broader domain of **LLM-Agent research**, not just "agentic research on code generation" claimed in your response, which is only a subset of this domain. You can refer to this paper list for this field, https://github.com/WooooDyy/LLM-Agent-Paper-List.
>
> ## Concerning the "REAL WORLD PHYSICS REASONING ABILITY" Claim
>
> Your response does not fully address my concern: does your benchmark and evaluation methodology truly assess the "REAL WORLD PHYSICS REASONING ABILITY" claimed in the title? As you mentioned, the problems in your benchmark are solvable using COMSOL alone. However, real-world physics reasoning typically exceeds what a single tool like COMSOL can achieve. It often involves integrating multiple tools, methodologies, and problem-solving techniques, such as combining FEA with analytical methods, probabilistic modeling, or other simulation tools like ANSYS or Abaqus.
>
> ## On the GUI vs. API/CLI Focus for COMSOL
>
> Your position does not justify entirely disregarding the GUI aspect of COMSOL in your benchmark. 1. The scenarios in SWE-bench and COMSOL differ fundamentally: while programmers often do not depend on a GUI for code-related tasks, engineers using COMSOL rely highly on its GUI for complex workflows. This distinction is critical. 2. Recent advancements in Vision-Language Models (VLMs) show significant progress in interacting with GUIs, as demonstrated by works such as CogAgent[1] and SeeClick[2]. Neglecting the GUI aspect limits the realism of your benchmark, as GUI interactions are integral to the real-world usage of COMSOL. If your rationale for excluding GUI interactions is that "LLMs struggle" with such tasks, it raises concerns about the completeness of your evaluation. By this logic, if LLMs perform poorly on your dataset, it might imply that LLMs should not be used for testing at all, which is a conclusion I don't believe you intend to make.
>
> [1] CogAgent: A Visual Language Model for GUI Agents, https://arxiv.org/abs/2312.08914
>
> [2] SeeClick: Harnessing GUI Grounding for Advanced Visual GUI Agents, https://arxiv.org/abs/2401.10935

---

> > ### Author Response · Authors · 2024-11-26
> > **Reply (1/2)**
> >
> > Thank you for your response. We reiterate that our intent was *neither* to claim that the building blocks for interacting with the Multiphysics API interface is the *only* possible realization of these blocks *nor* to introduce an agentic algorithm that is the best possible algorithm for this task, but instead to design components to faciliate LLM-API interaction for a baseline agent on this benchmark.
> > ## On LLM-Agent Research
> > With regard to your concern on connections with existing agentic work, we believe we have discussed generalist agentic frameworks and algorithms in our draft [Quotes 1, 2]. These include some of the papers at the link you provide.
> >
> > ## On using other software
> > "*As you mentioned, the problems in your benchmark are solvable using COMSOL alone.*"
> > It appears there might be a slight misunderstanding here. To reiterate Point 2 of our overall reply, the FEABench Gold problems do **not** preclude using other software. Since the FEABench Gold problems are physics / numerical problems, the Model Specifications field describes a physics / mathematical problem (example on Page 15) and the answer is **not** intended to be exclusively derivable via COMSOL Multiphysics®.
> >
> > ## On real-world physics reasoning ability
> > Our benchmark seeks to assess how well LLMs can represent complex objects and model interactions with physical phenomena to solve problems that require numerical (FE) analysis, and cannot typically be solved analytically alone -- a scenario encountered in diverse scientific and engineering contexts in the real world.
> >
> > We chose COMSOL Multiphysics® as a medium for this exploration to maximize coverage across physical phenomena, and use a single tool to solve problems involving electromagnetism, structural mechanics and heat transfer among others. Solving the problems in our benchmark requires making correct and consistent physics / engineering decisions and correctly generating code to steer the API to encode these choices. While we did not seek to exhaustively explore the heterogeneous landscape of other simulation tools and packages used across myriad subfields in physics and engineering, we respectfully contend that many of the **abilities** required to excel at using other tools (such as Ansys® or physics-subfield specific packages) are similar to those required in this context: making correct and mutually consistent physics decisions and learning to encode those decisions by calling tools or writing code to steer domain-specific software. Nonetheless, we have **updated our title**.
> > *On combining FEA with analytical methods:* We included an element of this direction in our agentic setup. The VerifierLLM component of our Evaluator sets an analytical estimate for the target description at the start of the agent experiment. Whenever the executability of the LLM's solution crosses 0.90, this LLM additionally compares the numerically-computed answer with its a priori analytical estimate, when providing its feedback.

---

> > > ### Author Response · Authors · 2024-11-26
> > > **Reply (2/2)**
> > >
> > > ## On GUI Integration
> > > Thank you for this suggestion. As we have discussed in our draft [Q1], leveraging developments such as those you cite to explore greater GUI integration would certainly be of interest to future work.
> > > *...does not justify entirely disregarding the GUI aspect:* Interaction with a GUI involves two components: 1) *receiving feedback* from the GUI and 2) *steering* the software using the GUI. We respectfully contend that we have *not* disregarded the first of these components. In Point 4 of our overall response we explained that there is a strong overlap between several aspects of the GUI experience and the information the agent receives as API feedback -- precisely one of the reasons we included the agent experiment. This is visible in the paper: the model tree in the left "Model Builder" panel in Figure 5 is shown to the LLM in the ToolLookupAgent as the 'Current Model Tree' (example on Page 18). The agent also receives linewise messages for the code it generated (Figure 2), much like a user would see either no messages when an action was correct, or a dialog box pop up for an incorrect action, for easier credit assignment of errors / inconsistencies. Indeed, our agentic blocks parsed this information as text, and the utility of adding visual snapshots of the GUI would certainly be an interesting future direction of exploration.
> > > *On difficulty of navigating the GUI:* Our point about the difficulty was *not* intended to eliminate the 2nd direction, i.e. operating the software using exclusively the GUI for future research. However, we believe this mode of interaction could make it more challenging to decouple the *source* of difficulty in this task. An additional question would now be whether the failure of the LLMs arises from their inability to correctly assimilate and interact with the GUI's visual elements vs making the right physics decisions and translating them to code. Ablating / decoupling these two challenges would revert to requiring a text-based representation of GUI information as we developed in this work as a baseline.
> > >
> > > **References:**
> > > **Quote[1], L520-523:** *It would be valuable to port blocks such as the Evaluator and the specialized
> > > functions into generalist agentic frameworks like AutoGPT and LangChain (Significant Gravitas; Chase, 2022) to explore possible performance gains and understand the optimum way to distil visual information from the GUI.*
> > > **Quote[2], L725-731:** *Other work in the LLM literature has focused on optimizing agent-tool call
> > > and design such as the ReAct and CodeAct strategies (Wang et al., 2024b; Yao et al., 2022). Beyond the realm of general-purpose programming, some works have sought to incorporate productivity APIs such as those for weather, email among others into agentic workflows (Qin et al., 2023; Basu et al., 2024). Our agentic approach shares similarities with the Reflexion strategy (Shinn et al., 2024), although in our case the Evaluator mainly returns subjective feedback from the API, and only queries its VerifierLLM when executability is already high.*
> > >
> > > We are grateful to you for your time and valuable comments.

---

> > > > ### Comment · Reviewer_oeq1 · 2024-11-28
> > > >
> > > > Thank you for your detailed response and clarifications.
> > > >
> > > > I understand your point that the FEABench Gold problems do not preclude using other software, and this does not contradict the statement that "these problems can be solved using COMSOL Multiphysics alone." This reflects a limitation of the benchmark itself, as the problems are still inherently tied to a single tool.
> > > >
> > > > Regarding the discussion on GUI, I understand your argument that there is an overlap between GUI feedback and API feedback. However, this overlap does not reduce the significance of GUI-based tasks. For example, in tasks related to web automation, earlier research focused primarily on text-based approaches, but in recent years, there has been a significant shift toward GUI interaction. The same applies here: incorporating GUI interactions can provide a richer and more realistic evaluation of the agent's capabilities, especially on using COMSOL Multiphysics®.
> > > >
> > > > Given your additional experiments and revisions to the paper, I raise my score from 3 to 5. Thank you for your efforts in addressing these concerns.

---

> > > > > ### Author Response · Authors · 2024-12-04
> > > > > **Reply**
> > > > >
> > > > > Thank you for your response and reassessment. We appreciate your valuable suggestions that helped us enhance our work.

---

### Official Review · Reviewer_Yw8s · 2024-10-20

**Soundness:** 2
**Presentation:** 2
**Contribution:** 3
**Rating:** 5
**Confidence:** 3

**Summary:**

The paper titled **"FEABench: Evaluating Language Models on Real World Physics Reasoning Ability"** introduces a novel benchmark for assessing the capability of large language models (LLMs) and LLM-based agents to perform finite element analysis (FEA) tasks. FEA is essential in many engineering domains for solving complex physical problems using numerical solvers. The paper tests LLMs' ability to read problem descriptions, reason over them, and interact with FEA software (COMSOL Multiphysics). It presents a benchmark called FEABench Gold, consisting of 15 manually verified FEA problems, and evaluates the performance of various state-of-the-art LLMs, such as Claude-3.5, GPT-4, and Gemini-1.5, in solving these tasks. The best strategy involved LLM agents capable of generating executable API calls for COMSOL. However, even the best-performing models struggled to solve any of the benchmark problems completely and correctly.

**Strengths:**

- The paper presents a novel benchmark focused on a crucial real-world domain (FEA) that has been underexplored in LLM evaluation. By requiring end-to-end reasoning from natural language to executable code for complex physics simulations, FEABench pushes the boundaries of what is currently possible with LLMs.

- The authors have taken care to ensure the problems are solvable, self-contained, and quantitatively verifiable. The evaluation framework is comprehensive, with multiple metrics capturing different aspects of solution quality.
- The inclusion of concrete examples (e.g., problem descriptions, code snippets) aids understanding.

- FEABench addresses an important gap in LLM evaluation by focusing on complex engineering problems that require both high-level reasoning and low-level code generation. Success on this benchmark could have major implications for automating engineering workflows. The multi-turn agent design also provides a valuable template for future work on LLM-based problem-solving systems.

**Weaknesses:**

- **Lack of Comparison with Human Performance**: Including a baseline comparison of human performance on these FEA tasks would significantly strengthen the evaluation. This comparison should involve both novice and expert COMSOL users to provide a spectrum of human capabilities.  Expert evaluation is particularly crucial to validate task difficulty,  establish performance benchmarks, assess solution quality and check benchmark integrity.

- **Limited Success of LLMs**: Despite the interesting problem posed, none of the LLMs tested were able to solve the benchmark problems completely and correctly. A detailed examination of where and why LLMs fail could provide valuable insights, potentially revealing common patterns in errors across different models or problem types. It would be beneficial to discuss how current LLM architectures might be fundamentally limited for these tasks and whether specialized components for mathematical reasoning or code generation could improve performance.

- **Lack of LLMs Candidates**: To provide a more comprehensive evaluation of current LLM capabilities, it would be beneficial to expand the range of models tested. This should include open-source model families such as LLaMA, Mistral, and Qwen. Testing multiple scales within each model family could reveal how model size affects performance on FEABench.  If possible, evaluating domain-specific models trained on scientific or engineering corpora would add further depth to the analysis. This broader evaluation would provide a more complete picture of the current state of LLM capabilities on FEA tasks and could uncover interesting differences in how various model architectures and training approaches handle these challenges.

- **Complexity of Task**: The paper could explore more how different levels of task complexity (e.g., simpler geometry or fewer physics variables) influence LLM performance. Additionally, the role of visual information from GUI could be better integrated into the evaluation process.



- **Not So Good Presentation**: The paper's presentation and structure could be improved to enhance clarity and readability.

  - As a benchmark paper, it fails to provide a clear, concise overview of the datasets' composition and size. The distinction between FEABench Gold and FEABench Large is not well explained, leaving readers confused about the exact number of problems in each dataset and the rationale behind this split. A clear statement of the dataset sizes (15 problems in FEABench Gold and 120 in FEABench Large) early in the paper, along with a brief explanation of why two separate datasets were created, would significantly improve understanding.

  - The evaluation metrics section is another area that lacks clarity. Some metrics, such as the Code Similarity Score, are introduced without sufficient justification or explanation of their relevance. The paper acknowledges that "two different code blocks could generate equivalent model subtrees," which raises questions about the utility of this metric in evaluating solution quality. The authors should either provide a stronger rationale for including this metric or consider removing it if it doesn't contribute meaningfully to the evaluation. Furthermore, the paper fails to effectively communicate the motivation behind introducing domain-specific metrics. While these metrics may be crucial for evaluating FEA solutions, the paper doesn't clearly articulate why existing general code evaluation metrics are insufficient and how the new metrics address these shortcomings. A more detailed explanation of how these domain-specific metrics capture important aspects of FEA problem-solving that general metrics miss would strengthen the paper's contribution.

**Questions:**

- Exploring the impact of training data and fine-tuning strategies is crucial. Current LLMs may not have adequate coverage of FEA-related knowledge in their training data, and domain-specific fine-tuning or data augmentation could potentially yield significant improvements. Additionally, discussing the potential of hybrid approaches that combine LLMs with other AI techniques, such as symbolic reasoning systems or neural-symbolic methods, could provide valuable insights into addressing the observed limitations.

- Could the authors clarify the main factors that contributed to the models’ failure to solve any benchmark problems completely? Was it primarily API interaction errors, poor physical reasoning, or something else?
- Have any attempts been made to include human feedback in the agent loops (e.g., human-in-the-loop evaluation)? If so, how does that impact the success of the LLM agents?
- Given the difficulty of the FEA tasks, what improvements to LLM architectures or agent designs do the authors suggest to improve their performance in future iterations?
- Could the authors include specific examples from the FEABench dataset to enhance the clarity and understanding of the benchmark? Providing concrete problem examples, along with their corresponding model specifications and expected outputs, would offer valuable insight into the types of tasks the LLMs are expected to handle.

---

> ### Author Response · Authors · 2024-11-23
> **Response (1/2)**
>
> We are grateful to you for your comprehensive review and appreciate your recognition of the potential relevance of our work.
>
> *Lack of Comparison with Human Performance:*
> In the status quo we manually verified that the answers to the FEABench Gold problems could be computed using the software and that the problem inputs were self-contained while generating / specifying the problem statements. We concur that an extensive comparative benchmarking with humans possessing various degrees of expertise would be of interest. A confounding factor in such a comparison would be the different modalities in which humans / LLMs are likely most adept at interacting with the software, since many users will likely exclusively use the GUI.
>
> *Limited Success of LLMs:*
> * Source of Difficulty – Skills: We introduced the ModelSpecs and Plan versions of the task on FEABench Gold to understand whether giving an explicit natural language (NL) plan enables the LLMs to solve the problem. The Plan2Code version of the tasks probes specifically the ability to *translate* predefined explicit NL instructions to code. The ModelSpecs2Code task requires the ability to make correct engineering decisions to represent the physics / geometry *and* translate it into code. A hypothetical LLM that excelled at translating the NL instructions to code but made poor engineering decisions would find the Plan2Code task significantly easier than the ModelSpecs2Code task. The observation that the LLM does not perform significantly better on the Plan2Code task than on the ModelSpecs2Code task indicates that translating NL instructions encoding the decisions to code is also a significant bottleneck.
> * Source of Difficulty–Across Problem Types: Figure 4 examined blockwise executability across the different conceptual substeps that comprise each FEA problem. The physics block has the lowest executability upon a single query, which motivates our focus on measuring performance on the physics steps. Figure 6 generates the same plot, contrasting blockwise executability across the “best” states after the agentic interaction with the initial distribution.
>
> **Additions:** For greater clarity, we provide a detailed qualitative comparison between the LLM’s solution and a ground truth solution in Appendix F and Figure 8\.
> The ability of alternative architectures or approaches at excelling at this task is an intriguing prospect: we have further discussed our bottlenecks in the Discussion and possible directions of exploration that might mitigate them. Approaches designed to boost the performance of LLMs on low-resource languages (languages without heavy representation in training data) and fine-tune on larger context lengths will likely benefit this domain.
>
> *Lack of LLMs Candidates:*
> We address this in Point 1 of the Overall Response. CodeGemma-7B-IT, in particular, was trained on an additional 500 billion tokens that included mathematics datasets and code repositories ([1]).
>
> *Complexity of Task:*
> * Task Complexity: One uniform way of probing task complexity across problems involves decoupling the ModelSpecs and Plan tasks.
> * While it would be interesting to explore other dimensions affecting complexity, given the multistep nature of the problem, other ways of slicing up the problem might not control for other variables that affect difficulty.
> * GUI Integration: Discussed in Point 3 of the Overall Response
>
> **References:**
> [1]: CodeGemma Team 2024. CodeGemma: Open Code Models Based on Gemma
>  arXiv:2406.11409

---

> > ### Author Response · Authors · 2024-11-23
> > **Response (2/2)**
> >
> > *Presentation:*
> > We appreciate these suggestions and have updated the structure to enhance readability.
> >
> > * L75-76: Clarifies the difference in FEABench Gold (manually verified) vs Large (algorithmically parsed). The dataset section, L151, also explains that while the Gold set commands the LLM to compute a single numerical value, the Large set does not possess a similar unique “answer”, due to being algorithmically parsed. The Large set offers a stronger statistical signal on model performance, while also being of value to fine-tuning experiments.
> > * Thank you for your feedback on this – we only intended to include code similarity as a “baseline code evaluation” metric as indicated in L181-184 (*We mainly report...metric across experiments*).
> > * There are 3 main motivations behind introducing the domain-specific metrics:
> >   (1) The presence of boilerplate code would confound metrics that act directly on “code string” space, and deem lines more similar than they actually are in terms of their execution simply by the presence of the “model.component…. “ initial prefix, and the note about different subblocks generating equivalent trees. This is also noticeable in the lack of significant variation of this metric across experiments.
> >   (2) Figure 4 demonstrates that the physics block is the one that suffers from the lowest executability – which motivates our focus on the physics metrics.
> >   (3) The poor executability of current LLMs also precludes meaningful use of pass@k / solve@k metrics. We have added this to Section 3 (Evaluation).
> >
> > **Addition:** Appendix F adds an illustrative example of how the physics metrics act upon a single example of an LLM solution paired with a ground truth solution. The preponderance of boilerplate code is evident here. We further modified the introduction of the Code Similarity Metric in Section 3, to more explicitly state our intent of introducing this metric only as a baseline measure of code similarity.
> > **Questions**
> >
> > * *Exploring the impact of training data and fine-tuning strategies is crucial.*
> >
> > **We added this experiment (Point 4 of the overall response).**
> >
> > * *Could the authors clarify the main factors…?*
> >
> > Solving the problems in our benchmark requires synthesizing all multiple skills–making sensible and consistent spatial / physics reasoning decisions, correctly following instructions, and translating these into code.
> > The fact that the Plan task is also challenging disproves the hypothesis that making the correct reasoning decisions is the only challenge – i.e. a well-specified plan alone is not sufficient to enable the LLMs to solve the problems.
> >
> > * *Have any attempts…?*
> >
> > We did not explore a human-in-the-loop setting. That would be a direction of interest to explore, with suitable control over the expertise of the humans involved.
> >
> > * *Given the difficulty of the FEA tasks…?*
> >
> > We **added** a discussion on these directions in Paragraph 2 of the Discussion.
> >
> > * *Could the authors include ...*
> >
> > In addition to the problem inputs in Appendix B, we additionally add a qualitative comparison of the LLM’s solution with the GT solution for the same problem in Appendix F.
> >
> > We express our gratitude for your detailed critique of our work and in helping us improve our draft. We hope we have answered your queries to your satisfaction, and if so, would appreciate it if you would consider re-evaluating our work.

---

> > > ### Comment · Reviewer_Yw8s · 2024-11-29
> > >
> > > Thank you for your detailed response and updates to the manuscript. I maintain my initial scoring due to the following key concerns:
> > >
> > > 1. While the response highlights logistical challenges, the absence of even a basic comparison with human performance, particularly with varying expertise levels, limits the evaluation's contextual relevance and benchmark reliability.
> > > 2. The inclusion of CodeGemma is noted, but the evaluation lacks breadth by excluding other prominent open-source models (e.g., LLaMA, Mistral). A more diverse set of models is crucial for a comprehensive understanding of LLM capabilities in this domain.
> > > 3. Despite revisions, the justification for certain metrics like Code Similarity remains insufficient. The utility and relevance of these metrics in evaluating FEA solutions are unclear, which weakens the evaluation framework.
> > > 4. The absence of experiments exploring human-in-the-loop setups and detailed analysis of how varying task complexity affects performance are missed opportunities to deepen the insights and practical relevance of the work.
> > >
> > > These issues are critical to the overall contribution and clarity of the paper.

---

> > > > ### Author Response · Authors · 2024-12-04
> > > > **Reply**
> > > >
> > > > We sincerely appreciate the time and effort you spent reviewing our work.
> > > >
> > > > Running controlled human-in-the-loop experiments and benchmarking performance against users of varying expertise would indeed be interesting directions for future exploration but require considerable effort beyond the discussion period. We added the following clarification to explain why we report code similarity in addition to the additional metrics we introduced: *We mainly report this metric as a baseline measure of code similarity, and to motivate our introduction of domain-specific metrics. The preponderance of boilerplate syntax, along with the fact that two different code blocks could generate equivalent model subtrees, are factors that contribute to the lack of meaningful variation of this metric across experiments.*
> > > >
> > > > On task complexity, a uniform way to slice up complexity that applies across tasks is the introduction of the Plan / ModelSpecs versions for each FEABench Gold problem, since all problems can be formulated with these 2 versions. Other ways of dividing problems might introduce other sources of complexity. For example, in the status quo, the Mathematics ODE problems may often lack a geometry or possess a simpler geometry, but their physics section might require specifying formulae that could be more non-trivial than, say, specifying a temperature boundary condition on an object composed of more geometric blocks.
> > > >
> > > > We reiterate our gratitude for your detailed review of our work and for helping us enhance our draft with additional analysis and  experiments on open-weights models and a fine-tuned model.

---

### Official Review · Reviewer_Lxi5 · 2024-11-01

**Soundness:** 3
**Presentation:** 3
**Contribution:** 1
**Rating:** 3
**Confidence:** 3

**Summary:**

The FEABENCH paper measures how well an LLM can solve a problem that requires doing “FEA = finite element analysis” by creating a dataset of problems to solve. Solving an FEA problem requires calling a tool through a series of API calls to describe the problem, the COMSOL Multiphysics tool is used in this paper and the tool does the numerical computation to solve the problem described to it through the API calls. The LLM must map an english problem description to a series of API calls to be processed/executed by the COMSOL Multiphysics tool to produce the solution. The paper measures the accuracy of a solution across many different metrics like the executability of the API calls produced, the correctness of the model tree produced, and the correctness of the target value returned.

**Strengths:**

Solving difficult FEA problems is an interesting and challenging domain for an LLM to master, it’s good to keep pushing LLMs to more difficult problems and see where they break, and what are the limits.

**Weaknesses:**

W0: Having to get a COMSOL Multiphysics R license to run the benchmarks sounded quite expensive and problematic for folks to get that expense approved in most schools and companies, so that is why I said poor for Contribution because I think students/researchers will pick other benchmarks that are cheaper to work on improving, with easier more direct solutions to measure against. I feel like a very niche set of folks would want to bite into this benchmark, with the cost to buy the tool, and the python wrapper complexities and tedious boiler plate interface to describe problems.

W1: It feels like this is a bit of niche problem domain, so the general Web data a model is trained on wouldn’t help prepare the model super well to use the FEA tool and call it with the appropriate sequence of API calls to get the correct answer.  But if the model was fine tuned on a bunch of problems/answers then maybe it could do a ton better. Would be interesting to know if finetuning on more real or synthetic data would help, or how using many more few shot examples in the prompt would help. It wasn’t clear to me how much documentation and examples the LLM was given to solve each problem - I thought it said 1 shot.

W2: The benchmark is so difficult that no LLMs can solve any of the problems completely. It seems to me the benchmark should include some super easy problems so something is solvable. But I guess that is what all the other metrics provide to some extent.

W3: I’m not an FEA expert but it feels like problems in the FEA domain are rather difficult and it feels like the ability to score well on this benchmark requires the model to do well in understanding the physics of the problem, and then also the LLM’s ability to program to the FEA solver’s API to describe the problem properly requires a lot of ugly boilerplate code it appears. So it feel like it would be hard to see clearly where the model was messing up, in understanding the physics or in understanding how to turn the physics into code that reflects the LLM’s understanding.

**Questions:**

About how much did it cost in LLM $$$ to run the evaluations across the different LLM models and how long did it take the LLMs?  Of the run time was the FEA software run time significant in the Golden dataset?

---

> ### Author Response · Authors · 2024-11-23
> **Response (1/2)**
>
> We are grateful for your thoughtful review of our paper.
> *W0: Having to get …to describe problems.*
>
> 1. Addressed in Point 2 of the Overall Response
> 2. Although several universities and engineering organizations have academic licenses for their students / staff to use COMSOL Multiphysics ®, we acknowledge and are cognizant of barriers that can impede accessibility. Unfortunately, many commercial-grade softwares require paid licenses. Eschewing widely-used FEA softwares would not provide the LLMs with the most authentic mirror of the environments FEA engineers use.
> 3. We appreciate your thoughts on this concern, and **ran a baseline attempt to query the LLMs to solve the FEABench Gold problems by generating Python code using Python numerical packages and FEniCS**. We used the \* same \* problem description as for the COMSOL Multiphysics ® experiments. The metrics we can use for this baseline are only whether a single sample of the code (without iterative interaction) was executable and the target relative error if a value was computed. **Only one generated code was executable, although the answer it computed had a greater than 100% relative error with respect to the correct answer.**
>
> *W1: It feels like … I thought it said 1 shot.*
>
> In terms of adding increasing amounts of documentation, we ran 3 versions of experiments on FEABench Gold for task ModelSpecs:
>
> * One-Shot (LLM sees a single example of problem \-\> code in-context)
> * One-Shot \+ Physics Documentation In-Context (the LLM sees the list of valid physics API options for interfaces and features \* in addition to \* the One-Shot example in-context)
> * Multi-Turn Agent: Here we first curated an LLM-annotated corpus of (NL Annotation \-\> Code Snippet) pairs, decomposed by the block of code. One of the tools in the agent enables the LLM to phrase a natural language query for a directed step (eg: “define an axisymmetric geometry”), and the retriever returns the 3 closest examples of (NL, Code) pairs. This is added to the context of the CorrectorAgent in the Agent experiment that can then use these examples of correctly formatted code to return the next solution.
>
> **Additions:**
> 1. We have restructured the presentation in Section 5 to make the last of these components more clear.
> 2. **Does fine-tuning boost performance**: Added in Point 4 of the overall response and Appendix G.
>
>  *W2: The benchmark … to some extent.*
>
> While a completely unsolvable benchmark may seem harder to track incremental progress on, we introduced the multifaceted evaluation scheme precisely as a way to assign ‘partial credit’ to solutions and glean signals of improvement. We find that our baseline agent is able to increase its ability to generate executable code over the course of a single experiment of 15 minutes.
>
> Moreover, considering the phenomenal pace of developments in LLM and agentic research, we contend that the (current) difficulty is an advantage, since an easier benchmark is more likely to be saturated soon after it is released. SWE-bench ([1]), one of the leading benchmarks spurring LLM development today, only had 1.96% of problems solvable when it was released a year ago.

---

> ### Author Response · Authors · 2024-11-23
> **Response (2/2)**
>
> *W3: I’m not an … the LLM’s understanding.*
>
> That is indeed an important question, and one that we have probed through multiple angles.
>
> 1. ModelSpecs vs Plan Tasks: We introduced the two versions of the tasks in FEABench Gold to understand precisely this question.  The ModelSpecs2Code task requires the ability to *both “understand the physics”* and make correct engineering decisions to represent it as well as translate it into code. The Plan2Code version of the tasks probes specifically the ability of “*understanding how to turn the physics into code”* i.e. Plan2Code requires the ability to correctly translate explicit natural language descriptions into code. The observation that the LLMs do not significantly outperform on the Plan2Code tasks than on the ModelSpecs2Code tasks indicates that translating correct decisions to code is also a significant bottleneck.
> 2. Evaluation: The multipronged evaluation scheme also serves to decouple different aspects of how close the LLM is to solving the problem. For example, a model with 100% executable code might not be generating code that is *relevant* to solving the problem. Thus the Model Tree Score measures the alignment of the LLM solution path (tree) with that of a ground truth solution path.
> 3. Boilerplate code: We show a qualitative example of the code generated by an LLM and a ground truth solution in Appendix E. The 3 SOTA LLMs we tested are able to adhere to the overall “boilerplate code” structure in the ground truth solution, and it is the options / arguments they pass to these calls that differ. The physics metrics also evaluate the *options within* the boilerplate calls and this example analyzes how the metrics flag inconsistencies for this example.
>
> *Cost and Runtime:*
>
> The agent experiment took around $2 per problem (as an approximate upper bound). Our agent experiments take around slightly over 12 minutes (between \~7-17 minutes) on average per problem. The FEA runtime is only a minuscule fraction of this time: parsing the LLM reply, evaluating it by executing it in COMSOL Multiphysics® and retrieving API messages took around 0.9-1.5s for a single LLM reply. **We added these statistics to Appendix E.**
>
> We hope we have addressed your questions in our revision and respectfully hope you might consider re-evaluating our work.
>
> **References:**
> [1]: Jimenez et al 2023. SWE-bench: Can Language Models Resolve Real-World GitHub Issues? arXiv:2310.06770

---

> > ### Author Response · Authors · 2024-12-04
> > **Reply**
> >
> > Thank you for your suggestions that helped us improve our work. To summarize, we hope we have addressed the concerns you highlighted to your satisfaction: by adding a fine-tuning experiment (W1), clarifying the three "levels" of documentation / examples incorporated in different experiments (W1), reiterating how our creation of two task versions for each problem in the Gold set serve to decouple the source of difficulty (W3) and reporting estimates of the cost and time distribution of the agent experiment (Q). One of our motivations for using our evaluation scheme was to glean some signal on different aspects of partial correctness instead of just receiving a binary "unsolved" score across LLMs and experiments (W2).

---

### Official Review · Reviewer_35U5 · 2024-11-04

**Soundness:** 2
**Presentation:** 3
**Contribution:** 2
**Rating:** 5
**Confidence:** 4

**Summary:**

This paper introduces FEABench, a benchmark for evaluating large language models (LLMs) in physics, mathematics, and engineering tasks via finite element analysis (FEA), with COMSOL Multiphysics as the selected software. The authors point to limited research on LLMs handling complex numerical simulations crucial in these fields.

FEABench includes two datasets:
- FEABench Gold with 15 human-verified problems offering quantitative targets, and
- FEABench Large with 200 parsed problems from COMSOL tutorials, which often require plot generation or multi-value computations rather than single, verifiable outcomes.

The study observes that while LLMs generate executable code, they face challenges like selecting appropriate physics interfaces and avoiding inaccuracies. To address this, the authors designed a multi-turn LLM agent system to iteratively refine solutions via COMSOL API interactions, feedback, and specialized sub-agents. This agent includes: an algorithmic call sequence to reduce errors, combined LLM and API feedback for consistency checks, and methods to sustain correction attempts despite tool call failures. The agent system consists of sub-agents, such as a Controller for solution generation, an Evaluator for feedback, a Corrector for solution adjustments, and a Tool Lookup Agent for retrieving information.

Despite significant progress in execution accuracy, complete and correct solutions to benchmark problems are still to be achieved.

**Strengths:**

- Challenging Benchmark: FEABench introduces a novel and complex benchmark for assessing LLMs in physics, mathematics, and engineering tasks using FEA software, offering a way to evaluate these models in real-world problem-solving scenarios.
- Evaluation: FEABench goes beyond correctness checks by employing a multi-dimensional evaluation strategy, providing a more detailed view of LLM capabilities.
- Agent-Based Framework: The multi-turn agent approach illustrates the potential of combining LLMs with feedback systems and tool integration to tackle complex, iterative tasks more effectively. The agent reaches an executability score of 0.88 (88% of the generated code lines are executable). This is considerably higher than the single-shot approaches. The agent shows substantial improvements in interface factuality, interface recall, and feature recall, indicating a better understanding and use of physics concepts in the code. The agent successfully computes a "Valid Target" in two out of fifteen problems.

**Weaknesses:**

- Limited Success: While the agent framework in FEABench improves code executability, this is not sufficient for overall benchmark success. While the agent computes valid target values in two problems, only one of those values falls within 10% of the ground truth answer. This is a crucial metric for determining true problem-solving success. Table 5 shows that the agent-based approach does not lead to significant improvements in code similarity compared to single-shot prompts. The agent showed only modest improvement in accurately setting properties of physics features, and it failed to fully solve any FEABench problem. This raises questions about whether the agent's search strategy is well-suited for tackling the benchmark's challenges.
- Absence of Agent Baselines: Although the paper proposes an agent-based approach, it does not provide a comparative analysis against existing (rich set of) LLM agent frameworks, making it difficult to assess improvements in code generation or overall performance. As a result, the primary contribution seems to be limited to the introduction of the FEABench dataset.
- The iterative nature of the agent-based approach involves multiple cycles of code generation, execution, feedback analysis, and code correction. This process can be computationally expensive, particularly when dealing with complex FEA problems that require significant computational resources for simulation.

**Questions:**

I would appreciate if the authors could elaborate on the novelty of their agent design, especially in comparison to the extensive existing literature on LLM agent development. Specifically, clarifying how their approach differs in structure, feedback mechanisms, or task handling would help understand the potential algorithmic advancements introduced in this work.

---

> ### Author Response · Authors · 2024-11-23
> **Response (1/2)**
>
> We thank you for your valuable feedback and appreciate your recognition of the utility of our evaluation metrics.
> **Changes made in response to your feedback:**
>
> - Comparative Analysis and Agent Design: We reframed Section 5, 7 and the Related Work Appendix, and strengthened the ways in which we are related to existing agentic approaches in the literature as well as distinguish ourselves from them in Related Work (discussion on Reflexion), and the Discussion.
> - Structure: Separated the elements related to the Multiphysics-Software Interface from the rest of the agent control flow
> - Feedback: Specific point under Design Elements in Section 5
> - Task Handling: Discussed in Section 5.1. (To minimize…input context)
> - Unique aspects of our design: Described in Design Elements.
>
> **Response:**
>
> *Limited Success:*
> * Our objective behind introducing the agent was to devise a best-effort baseline that seeks to address obvious sources of difficulty in the single-turn setup, such as the absence of interaction with and knowledge about the API. We find that in spite of curating such an environment, the problems remain a challenge. Our objective was not to show that our existing agentic setup is capable of completely solving the benchmark, but rather to design an LLM-Agentic interface for this domain that finds a path toward mitigating some of the issues in the single-turn setting, as is observed in the improved executability-focused metrics.
>   Moreover, if the agent was, in fact, able to perfectly solve these problems, this would indicate that the benchmark would be easier to solve, and would thus pose less of an “unsolved” challenge to the community.
>   **Changes:** A side note about the lack of improvement in “code similarity” in particular: As indicated in the evaluation section (L174-175 in the submitted draft), we mainly chose to include code similarity score as a baseline metric for code similarity. The equivalence between multiple blocks of code and the large amount of boilerplate code dilutes the use of this metric and it has very little variation across all experiments, and not just the agent. We have clarified our description of Code Similarity in the revised version to address this.
>   **L181-184:** *“We mainly report this metric as a baseline measure of code similarity, and to motivate our introduction of domain-specific metrics. The preponderance of boilerplate syntax, along with the fact that two different code blocks could generate equivalent model subtrees, are factors that contribute to the lack of meaningful variation of this metrics across experiments.”*
>
> *Absence of Agent Baselines:*
>
> To the best of our understanding, SOTA agent frameworks cannot directly be used off-the-shelf in an optimum fashion for our software setting. Thus a direct quantitative analysis would not be an apples-to-apples comparison. The recent landscape of agent research has focused on code generation in general-purpose languages such as Python which often requires mastery over skills like localizing and fixing bugs across multiple files. For example, the key components frameworks like [1], [2], [3] that excel at SWE-Bench ([4]) enhance utilities for file manipulation and leveraging tools from existing Python packages. These utilities are optimized for software engineering workflows but are less relevant to FEA workflows. As a concrete example, solving an FEA problem in FEABench Gold does not require the user / LLM to read in files or navigate changes to pre-existing code across multiple modules or directories.
>
> Where relevant, however, there are *conceptual* elements that we design for our Agent-Software interface that are shared with other general-purpose agents. Other works ([1]) have used the ability to make specialized tool calls . In our interface, the ToolLookupAgent serves this purpose, as it uses the Langfun interface to call different specialized Python functions that are **relevant** to interacting with the COMSOL Multiphysics(R) API, such as QueryModelTreeProperties.
>
> Previous work on building agent-software interfaces has underscored the need to carefully design interfaces that are well-adapted to the problem domain and actions that provide feedback in a concise yet information-dense fashion and are cognizant of LLM limitations. We sought to adhere to these principles in designing our agent-software interface tailored to enabling an LLM to interact with an FEA software.
>
> **References:**
> [1]: Wang et al 2024. OpenHands: An Open Platform for AI Software Developers as Generalist Agents. arXiv: 2407.16741
> [2]: Wang et al 2024. Executable Code Actions Elicit Better LLM Agents. arXiv: 2402.01030
> [3]: Yang et al 2024. SWE-agent: Agent-Computer Interfaces Enable Automated Software Engineering. arXiv: 2405.15793
> [4]: Jimenez et al 2023. SWE-bench: Can Language Models Resolve Real-World GitHub Issues? arXiv:2310.06770

---

> ### Author Response · Authors · 2024-11-23
> **Response (2/2)**
>
> *The iterative nature … for simulation.*
>
> In our agentic experiments, parsing the LLM reply, evaluating it by executing it in COMSOL Multiphysics® and retrieving API messages took around 0.9-1.5s for a single LLM reply. Our agent experiments take around slightly over 12 minutes (ranging from 7-17 minutes) on average per problem. Several challenging benchmarks, even in high-resource languages such as Python, have required iterative interaction to solve. Other work ([1])  that has focused on data science problems for example, the best performing agent is given 24 hours to solve a single problem. **Nonetheless, we added a Fine-Tuning experiment (Point 4 of the overall response), since that is likely a plausible route to enhance solvability without multiple iterations.**
>
> *I would appreciate … in this work.*
> We hope we have addressed this in the changes made in the revision (changes made in response to your feedback). The unique elements we adopted for our setting include:
> - Analytical-Numerical Consistency: If the VerifierLLM component of the Evaluator is asked to critique the solution, it compares the LLM’s computed numerical answer with its analytical guess.
> - LLM-Assisted Semantic Code Search: Since our experiments demonstrate that at least part of the challenge is from the lack of familiarity of the LLM with the code snippets. We used an auxiliary LLM to annotate individual compositional blocks that comprise an entire code. While the *specific* corpus we use is COMSOL-specific, the approach is not, and can be generalized to other low-resource programming languages (LRPLs) or other domain-specific coding languages.
>
> We are grateful to you for your insights and feedback and have modified our draft accordingly.
>
> **References:**
> [1] Chan et al 2024. MLE-bench: Evaluating Machine Learning Agents on Machine Learning Engineering

---

> > ### Comment · Reviewer_35U5 · 2024-11-29
> >
> > Thank you for your response. I would like to increase the score to 5. I appreciate the efforts made to address a well-identified gap in the field of LLM reasoning for real-world physics. While the response provides valuable insights into agent design, my concern about the absence of agent baselines remains.  A benchmark and dataset paper should include extensive experiments that demonstrate how the dataset or benchmark enables meaningful comparisons and provides insights into model performance. This paper does not sufficiently explore how existing approaches in the current landscape of agent research for code generation would perform on this benchmark. Providing results from such baselines, even if they fail, could enhance the paper's value by clarifying the challenges and establishing the benchmark as a stepping stone for future research.

---

> > > ### Author Response · Authors · 2024-12-04
> > > **Reply**
> > >
> > > We are grateful for your effort in reviewing and helping us enhance our work. We immensely appreciate your acknowledgement of the gap our work sought to address and for your reevaluation. An extensive benchmarking of general purpose code generation agentic frameworks would be an interesting direction for future exploration however, robustly readapting these frameworks and replacing their components with the API interaction blocks we built here required effort beyond the rebuttal period.

---

### Author Response · Authors · 2024-11-23
**Overall Response (1/2)**

Dear Reviewers R3U5, Lxi5, Yws85, oeq1,

We are grateful to you for your thorough reviews of our paper and value your constructive suggestions and critique of our work. We seek to addressed shared comments here:

1. **Evaluating Open Source LLMs:** We added evaluations of the open source Gemma family on FEABench Gold – specifically evaluating CodeGemma-7B-IT, Gemma-2-9B-IT and Gemma-2-27B-IT.

| Experiment        | Executability   | Model Tree Score  | Code Similarity | Valid Target |
|-------------------|-----------------|-------------------|-----------------|--------------|
| Gemma-2-27B-IT    | 0.56±0.05       | 0.47±0.07         | 0.15±0.02       | 0/15         |
| Gemma-2-9B-IT     | 0.44±0.06       | 0.28±0.06         | 0.11±0.02       | 0/15         |
| CodeGemma-7B-IT   | 0.52±0.07       | 0.35±0.06         | 0.12±0.02       | 0/15         |


| Experiment        | Interface Factuality | Interface Recall | Feature Recall | Feature Property Recall | Feature Dimension           |
|-------------------|-----------------------|------------------|----------------|--------------------------|-----------------------------|
| Gemma-2-27B-IT    | 0.69±0.13            | 0.50±0.14        | 0.14±0.08      | 0.11±0.07                | -                           |
| Gemma-2-9B-IT     | 0.70±0.15            | 0.43±0.14        | 0.06±0.04      | 0.07±0.07                | -                           |
| CodeGemma-7B-IT   | 0.45±0.13            | 0.21±0.11        | 0.17±0.09      | 0.07±0.07                | -                           |




2. **FEABench Gold problems do not preclude using other software:** The FEABench Gold problems are physics / numerical problems. Their answers are \*not\* intended to be exclusively derivable via COMSOL Multiphysics®. This can be seen in the absence of software-specific references in the problem description in Figure 1\. We chose COMSOL Multiphysics® as the choice of software since it abstracts away several of the mathematical minutiae (like the variational formulation of the PDE) unlike FEniCs (a finite element method package in Python) and thus has a lower barrier of entry for a novice and can be used to solve problems involving much more non-trivial geometries than those in our benchmark.
3. **Leveraging GUI feedback:** While we do not explore the ability of LLMs to manipulate the GUI by clicking at correct locations, there is a strong overlap between the information we query from the API and the GUI graphics. In the context of the agent,
* API Messages: API messages corresponding to errors (red lines in Figure 2\) would otherwise pop up in dialog boxes to a user using the GUI. The agent receives this information from the Evaluator block (and sees past attempts’ errors in the Execution History).
* The Model Tree (see Section B.1.3) closely resembles the nested structure in the left “Model Builder” panel in Figure 5\. When calling tools, the LLM sees the *dynamic* state of the tree under the description of the QueryModelTreeProperties tool.
* The options in the Property-related dropdowns and boxes visible in the second panel in the GUI from the left (in Figure 5), are represented as a dictionary when the QueryModelTreeProperties tool is correctly called (see Figure 2, first ToolCall).

	In the context of evaluation (reply to Reviewer Yw8s):

* The Executability metric further measures the fraction of API messages parsed as “incorrect”.
* Model Tree Score: This measures the similarity between the tree in a correct solution with that in the LLM solution.

It would be interesting to explore alternative ways of integrating visual information, or whether directly manipulating the GUI could boost performance. However, given that (1) LLMs are primarily trained on text and code datasets, and (2) research [1],  that claim that LLMs struggle with directly interacting with visual interfaces such as IDEs relative to humans, it is possible that this could further compound the difficulty of the task.

**References:**
[1]: Jimenez et al 2023. SWE-bench: Can Language Models Resolve Real-World GitHub Issues? arXiv:2310.06770

---

> ### Author Response · Authors · 2024-11-23
> **Overall Response (2/2)**
>
> 4. **Fine-Tuning on FEABench Large:** We added an experiment in which we fine-tuned Gemini 1.5 Flash on a subset of FEABench Large using a zero-shot prompt, to adhere to context limits. We then examined the performance of the fine-tuned checkpoint on FEABench Gold and contrasted it with the performance of the untuned LLM and the untuned LLM with a One-Shot prompt. The fine-tuned LLM does not offer an advantage over the One-Shot prompting scenario. This is likely in large part due to token window limits during fine-tuning. We discuss this in more detail in Appendix G.
>
>
> | Description            | Executability          | Model Tree Score       | Code Similarity        | Valid Target |
> |------------------------|------------------------|------------------------|------------------------|--------------|
> | FT &#124; Zero-Shot    | 0.50±0.06             | 0.24±0.06             | 0.10±0.02             | 0/15         |
> | Baseline &#124; Zero-Shot | 0.08±0.03          | 0.16±0.05             | 0.08±0.01             | 0/15         |
> | Baseline &#124; One-Shot | **0.57**±0.04       | **0.49**±0.06         | **0.14**±0.02         | 0/15         |
>
>
> | Description            | Interface Factuality   | Interface Recall       | Feature Recall         | Feature Property Recall | Feature Dimension      |
> |------------------------|------------------------|------------------------|------------------------|-------------------------|------------------------|
> | FT &#124; Zero-Shot    | 0.42±0.15             | 0.36±0.13             | 0.13±0.09             | **0.15**±0.09          | -                     |
> | Baseline &#124; Zero-Shot | -                  | 0±0                   | 0.20±0.11             | 0.07±0.07              | -                     |
> | Baseline &#124; One-Shot | **0.80**±0.11       | **0.71**±0.13         | **0.36**±0.11         | 0.01±0.01              | **0.53**±0.18         |
>
> We have discussed your other comments in our responses below. We reiterate our gratitude for helping us improve our work.

---

### Meta-Review · Area_Chair_UuY9 · 2024-12-18

**Metareview:**

The paper investigates the ability of LLMs to perform finite element analysis in a tool use/agentic framework. Primary strength is that this is a novel and interesting domain, which is unfortunately connected to his primary weakness, which is that the application area is quite niche (certainly much more than the title would indicate). For that reason it's not clear that this would be a broad interest to the ICLR community, and so absent a much broader range of physical reasoning tasks (including commonsense/real world physical reasoning), this particular work is not ready for publication.

**Additional Comments On Reviewer Discussion:**

The authors raised a variety of valid points about the evaluation, such as the limited range of language models used, /but/ the authors did a great job of addressing that particular issue. They did not rebut the concern about the limited range of problems considered however.

---

### Decision · Program_Chairs · 2025-01-22

Reject